# Primary Care Professionals’ Acceptance of Medical Record-Based, Store and Forward Provider-to-Provider Telemedicine in Catalonia: Results of a Web-Based Survey

**DOI:** 10.3390/ijerph17114092

**Published:** 2020-06-08

**Authors:** Josep Vidal-Alaball, Francesc López Seguí, Josep Lluís Garcia Domingo, Gemma Flores Mateo, Gloria Sauch Valmaña, Anna Ruiz-Comellas, Francesc X Marín-Gomez, Francesc García Cuyàs

**Affiliations:** 1Unitat de Suport a la Recerca de la Catalunya Central, Fundació Institut Universitari per a la recerca a l’Atenció Primària de Salut Jordi Gol i Gurina, 08272 Sant Fruitós de Bages, Spain; gsauch.cc.ics@gencat.cat (G.S.V.); xmarin.cc.ics@gencat.cat (F.X.M.-G.); 2Health Promotion in Rural Areas Research Group, Gerència Territorial de la Catalunya Central, Institut Català de la Salut, 08272 Sant Fruitós de Bages, Spain; aruiz.cc.ics@gencat.cat; 3Department of Economics and Business, University of Vic-Central University of Catalonia, 08500 Vic, Spain; jlgarcia@uvic.cat; 4TIC Salut Social-Generalitat de Catalunya, 08005 Barcelona, Spain; flopez@ticsalutsocial.cat; 5CRES&CEXS-Pompeu Fabra University, 08003 Barcelona, Spain; 6Unitat d’anàlisi i qualitat. Xarxa Sanitària i Social de Santa Tecla, 43003 Tarragona, Spain; gfloresm@xarxatecla.cat; 7Centre d’Atenció Primària Sant Joan de Vilatorrada, Gerència Territorial de la Catalunya Central, Institut Català de la Salut, 08250 Sant Joan de Vilatorrada, Spain; 8Sant Joan de Déu Hospital, Catalan Ministry of Health, 08950 Esplugues de Llobregat, Spain; francesc.garcia@umedicina.cat

**Keywords:** telemedicine, primary health care, acceptability of health care, surveys and questionnaires

## Abstract

While telemedicine services enjoy a high acceptance among the public, evidence regarding clinician’s acceptance, a key factor for sustainable telemedicine services, is mixed. However, telemedicine is generally better accepted by both patients and professionals who live in rural areas, as it can save them significant time. The objective of this study is to assess the acceptance of medical record-based, store and forward provider-to-provider telemedicine among primary care professionals and to describe the factors which may determine their future use. This is an observational cross-sectional study using the Catalan version of the Health Optimum questionnaire; a technology acceptance model-based validated survey comprised of eight short questions. The online, voluntary response poll was sent to all 661 primary care professionals in 17 primary care teams that had potentially used the telemedicine services of the main primary care provider in Catalonia, in the Central Catalan Region. The majority of respondents rated the quality of telemedicine consultations as “Excellent” or “Good” (83%). However, nearly 60% stated that they sometimes had technical, organizational or other difficulties, which might affect the quality of care delivered. These negatively predicted their declared future use (*p* = 0.001). The quality of telemedicine services is perceived as good overall for all the parameters studied, especially among nurses. It is important that policymakers examine and provide solutions for the technical and organizational difficulties detected (e.g., by providing training), in order to ensure the use of these services in the future.

## 1. Introduction

Aside from emerging evidence on the clinical impact of telemedicine, which has shown that it can provide effective health services at a lower cost [1,2,3], telemedicine services enjoy a high degree of acceptance among the public, as studies show that it saves them significant time [4,5,6,7]. In addition, it has been shown that avoiding travel to health centers can reduce air pollution in the context of the current climate crisis [8,9]. While clinician acceptance is a key factor for sustainable telemedicine services [10,11,12], evidence is mixed. A comprehensive systematic review reported that clinicians were highly satisfied with both store-and-forward and real time telemedicine [13]. However, other evidence suggested a subtle/nuanced effect: although services are generally well accepted by professionals, they express concerns regarding the increased workload that this might entail [14,15]. Such concerns appear to be more prevalent among primary care physicians than among nursing staff, who also express a fear that telemedicine could potentially undermine their professional autonomy [16]. While some studies suggest that telemedicine is better accepted by patients who live in rural areas [17], professionals in these fields are also more satisfied with the service as it facilitates contact with hospital specialists, improving their professional development [18,19]. Other studies also show the benefits of telemedicine for professionals in terms of professional development [20,21], as well as suggesting that it enabled them to approach their patients with greater knowledge [22].

The Catalan Health Care System dispenses services for 7.6 inhabitants, providing universal coverage through a tax-based system. Administratively, it is composed by a single public payer and multiple service providers publicly or privately owned, with an integrated system. Some of its peculiarities are the role of community and primary health care and the increasing use of information technologies and digital health [23]. In Catalonia, telemedicine is used in numerous areas, such as screening, diagnosis and the treatment of disease, asynchronously in particular. The most widely used are interconsultations between primary and hospital care professionals and teleconsultations between primary care professionals and patients [24]. With regard to the former, evidence suggests that they are cost-effective, serve to reduce face-to-face visits and that the economic benefit is mainly enjoyed by the patient [2]. Regarding the latter, recent studies show that they can also be useful in reducing face-to-face visits mainly for administrative reasons [25,26]. If, as these experiences suggest, telemedicine is shown to be socially desirable, it is important that healthcare professionals (who are the ones who make the decision to use it) are satisfied with it, even though it may not be of any particular benefit to them, meaning that they are active promoters of such services.

Numerous questionnaires attempt to study the acceptance of healthcare professionals regarding telemedicine [27,28,29]. Among them is the validated Catalan version of the “Health Optimum Questionnaire” [30], inspired by the areas and simplicity of the technology acceptance model [31]. In this context, the objective of this study was to assess the acceptance of telemedicine services among primary care professionals in the Catalan central region using the aforementioned questionnaire, and to describe the factors which may determine the use of telemedicine services in the future.

## 2. Methods

### 2.1. Study Questionnaire

This is an observational cross-sectional study using the Catalan version of the Health Optimum questionnaire, a Technology Acceptance Model-based validated survey which aims to measure the degree of satisfaction with telemedicine services and describing the factors that can determine its future use [30]. The questionnaire, consisting of 8 short questions with 3 or 5 response options using a Likert scale, is based on the two main concepts of ease of use and comprises three dimensions of perceived usefulness: individual context, technological context and implementation or organizational context [31]. An internet polling tool was used to anonymously send the questionnaire to all primary health care professionals who potentially had contact with the four medical record-based, store and forward provider-to-provider telemedicine specialties (teledermatology, teleulcers, teleophthalmology and teleaudiometry) of the 17 primary health care teams in the counties of Bages, Moianès and Berguedà of the Catalan Central Region of the Catalan Health Institute (the main primary care provider in Catalonia). The poll was sent by the Institute’s Central Catalonia Research Unit to 661 healthcare professionals’ email addresses on 18 May 2018. A reminder was sent on 30 May and the questionnaire was definitively closed on 8 June 2018.

The first question asked whether the health professional had used a telemedicine service at any time. If the response was “Yes”, they continued with the questionnaire, while if the response was “No”, the questionnaire ended. The survey received a total of 163 responses (response rate: 24.7%). Those who stated that they had never used a telemedicine service (40) or did not complete the survey (15) were excluded from the analysis. Thus, the sample under analysis is comprised of 108 participants (Figure 1).

When numbers are small, Chi-square tests can give inaccurate results. This problem can be solved by grouping some of the answers. In questions 1 and 2, we grouped together the positive variables “excellent” and “good” on one hand and the negative variables “regular” and “bad” on the other. In question 3, “much better” was grouped with “better”, and “worse” with “much worse”. In question 4, “very comfortable” was grouped with “somewhat conformable”, and “somewhat uncomfortable” with “very uncomfortable”. The Chi-square was recalculated after grouping the variables and no differences were found in the results. 

Linear correlations between questions were calculated, in order to better describe possible relations and to try to summarize information for future studies on the quality of telemedicine. For this purpose, the 112 participants who replied to some of the questions were included. Only Pearson correlation coefficients (PCC) higher than 0.5 were considered. A multivariate linear regression model was used to assess which variables could predict the future use of telemedicine services.

The programs Epi InfoTM v7.2.2.1 (Centers for Disease Control and Prevention, Atlanta, GA, USA) and SPSS v23 (SPSS IBM Inc., Chicago, NY, USA) were used for statistical analyses. Results were considered significant with *p* < 0.05. The study protocol was approved by the University Institute for Primary Care Research (IDIAP) Jordi Gol Health Care Ethics Committee (Code P16/046).

### 2.2. Sample Characteristics

Besides the questionnaire, the poll collected additional information to supplement the data with background characteristics of the respondents (Table 1). The typical profile was a 48-year-old woman, a General Practitioner, who mostly used the teledermatology service. The respondents declared having used one or more telemedicine service a total of 1515 times during the previous year, an average of 14.02 (with a median of 5 and a mode of 3) per respondent.

## 3. Results

### 3.1. Survey Results

The main results of the questionnaire are shown above (Figure 2).

The majority of participants rated the quality of telemedicine consultations as excellent (65.18%) or good (17.86%). They also mostly rated the technical quality of telemedicine consultations as good (68.75%). When comparing the quality of care delivered by the telemedicine service with the quality of traditional care, nearly half of the respondents stated that the quality was about the same; around 30% considered the quality of care to be better or much better, and just 20% considered the quality of care to be worse.

Half of the participants responded that they feel as comfortable during the telemedicine consultation as in a face-to-face consultation, while the rest felt somewhat comfortable or very comfortable. More than 70% of respondents considered that telemedicine services could improve patients’ health status. However, nearly 60% of respondents stated that they sometimes had technical, organizational or other difficulties which might affect the quality of care delivered by telemedicine services.

Finally, when asked about future use of telemedicine services, nearly 70% of respondents wanted to continue to use the services in the same way, whilst the rest wanted to use the service but with some improvements. The mean and standard deviation of the responses are shown in Table 2.

### 3.2. Sensitivity Analysis.

We looked at differences in responses to the eight questions in the questionnaire according to professional categories, identifying significant statistical differences in three cases. First, with respect to question 3, nursing staff rated the quality of care delivered by the telemedicine services as significantly better compared with medical staff (*p* < 0.001). Second, regarding question 6, medical staff reported having experienced more technical difficulties than nursing staff (*p* < 0.05). Finally, with regard to question 7, medical staff stated having experienced more organizational and other difficulties that might have affected the quality of care delivered by the telemedicine services than nursing staff (*p* < 0.001) (Table 3). The differences in responses by gender and age have no statistical differences between groups.

Looking at differences in responses to the eight survey questions, in relation to the number of times respondents have used any of the various telemedicine services in the previous year, significant statistical differences were found in three questions. Respondents who used telemedicine services more often rated the quality of care of these services as significantly worse, compared with respondents who use them less often (Figure 3) (*p* < 0.05). They also stated having experienced more technical difficulties and more organizational and other difficulties compared with respondents that used telemedicine services less often (*p* < 0.001).

### 3.3. Multivariate Linear Regression Model

A positive correlation (PCC = 0.728) was found between the overall quality of the telemedicine consultation and the rating of the technical quality of said consultation, between the overall quality of the telemedicine consultation and the future use of telemedicine (PCC = 0.583), and between the technical quality of the telemedicine consultation and the future use of telemedicine (PCC = 0.505). A negative correlation (PCC = 0.531) was found between organizational and other difficulties that might affect the quality of care delivered by the telemedicine service and the future use of telemedicine. We found no correlation between the perceived quality of telemedicine rendered and the perceived barriers of use (Table 4).

The multivariate linear regression showed two variables to be good predictors for the future use of telemedicine services: the overall quality of the telemedicine consultation (*p* < 0.005) which positively affects the future use, and organizational (or other) difficulties that might affect the quality of care delivered by the telemedicine service (*p* < 0.001), which negatively impacts the future use (Table 5).

## 4. Discussion

In a public health system such as the one found in Catalonia, in which health professionals can choose between telemedicine and usual care (you cannot force them to use telemedicine), understanding the “drivers” of acceptance is essential, so that its use is widespread. The short questionnaire used in this study has allowed an easy, massive and real-time measurement of professional acceptance. If telemedicine is shown to be socially desirable, it is important that health professionals (who are the ones who make the decision to use it) are satisfied, although it may not be of particular benefit to them, so that they become active promoters. This study shows that, in general, the professionals in central Catalonia who responded to the survey are pleased with the telemedicine services available to them and that they will continue to use them as they are currently designed. They state that these services have an overall perceived quality similar to usual care; they do not show any anxiety when using them and believe that they can improve the health of their patients.

However, the Catalan public health system needs to improve in terms of the technical and organizational difficulties which the professionals claim to have suffered while using the services; key elements for their acceptance: since those who had difficulties would be more reluctant to use the services in the future. It is worth noting that none of the healthcare professionals received any training prior to the utilization of the telemedicine services, and nor did new health professionals who joined the primary care teams. If the healthcare professionals had received some form of training, they would probably have had fewer of the difficulties which negatively affect the use of the services. Furthermore, the results show that we can expect a high degree of acceptance by nurses in managing a systemic change towards models with more telemedicine services, unlike that which was shown in other studies [16].

Respondents who used the telemedicine services more often rated the quality of care of these services significantly lower than respondents who use them less often and reported having experienced more technical difficulties and more organizational and other difficulties, compared with respondents who used telemedicine less often. The fact that those who use the tools more are more critical is a wake-up call to technical managers of telemedicine services, suggesting that their experiences should be incorporated into the development of the technology.

### Limitations

In relation to the sample universe, there was an overall response rate of just 24.7% among all physicians, nurses and dentists from the counties of Bages, Moianès and Berguedà, regardless of whether they used a telemedicine service or not. However, the 108 participants can be seen as a representative sample of the professionals who use telemedicine in this area, since it is not expected that many more professionals use it, although the exact number has not been recorded. Considering that nurses make up almost half of the sample, the socio-demographic profile of the participants is quite similar to that shown by analyses of primary care professionals in Catalonia [32].

Using a web-based survey implies a possible bias in obtaining answers from those with better technology management. It is also possible, for example, that more people in favor of telemedicine services responded to the survey. Professionals who had negative experiences may have not responded to the questionnaire. Moreover, because the data are based on self-reported measurements, they are potentially highly correlated. It should also be borne in mind that this survey was conducted in a semi-rural context, where previous studies have shown that, in general, there is a greater acceptance of telemedicine [18,19]. In the future, it would be interesting to see if the same results are reproduced in an urban context. 

We are aware that the simplicity of the eight questions questionnaire might imply that the tool loses some capacity to capture specific insights. However, the questionnaire was specifically designed to be simple to use and was validated in a previously published study, showing a high reliability index [30].

We performed correlations using ordinal qualitative variables, as we thought that it could help us to understand relations among the factors we were dealing with, such as satisfaction, difficulties and future use. We also used a multivariate linear regression model as a qualitative exercise to propose possible relations between variables, and also in order to propose the key questions to measure the success of teledermatology. These results should not be interpreted as a basis on which to draw firm conclusions or certainties, or to quantify relations, but rather to observe possible relations and to find possible indicators of success. Finally, it should be mentioned that, in future studies, these results should be compared with other perspectives, such as users/patients or healthcare administrators.

## 5. Conclusions

We assessed the acceptance of telemedicine services amongst health professionals in the Catalan central region, using a short validated questionnaire. Results show that the quality of the services are overall perceived as good for all the parameters studied. Although no great enthusiasm was shown, the results are good, especially among nursing professionals. The healthcare professionals also reported that the technical quality is good, although they often experience technical and organizational problems. It is noteworthy that less than 1% intend to stop using telemedicine in the future. It is necessary that policymakers address the technical and organizational difficulties which professionals have encountered (e.g., by providing training), if they are to ensure the use of these services in the future.

## Figures and Tables

**Figure 1 ijerph-17-04092-f001:**
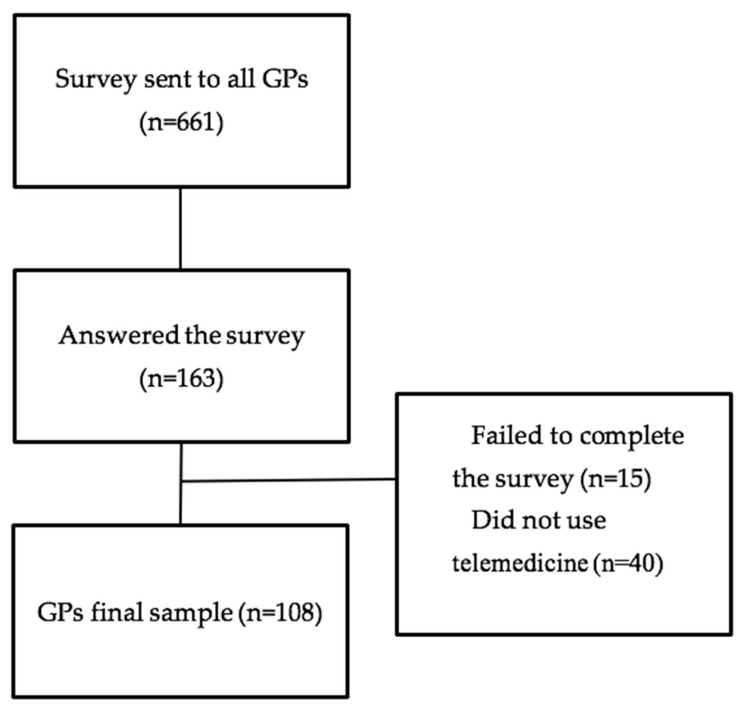
Participant Flow.

**Figure 2 ijerph-17-04092-f002:**
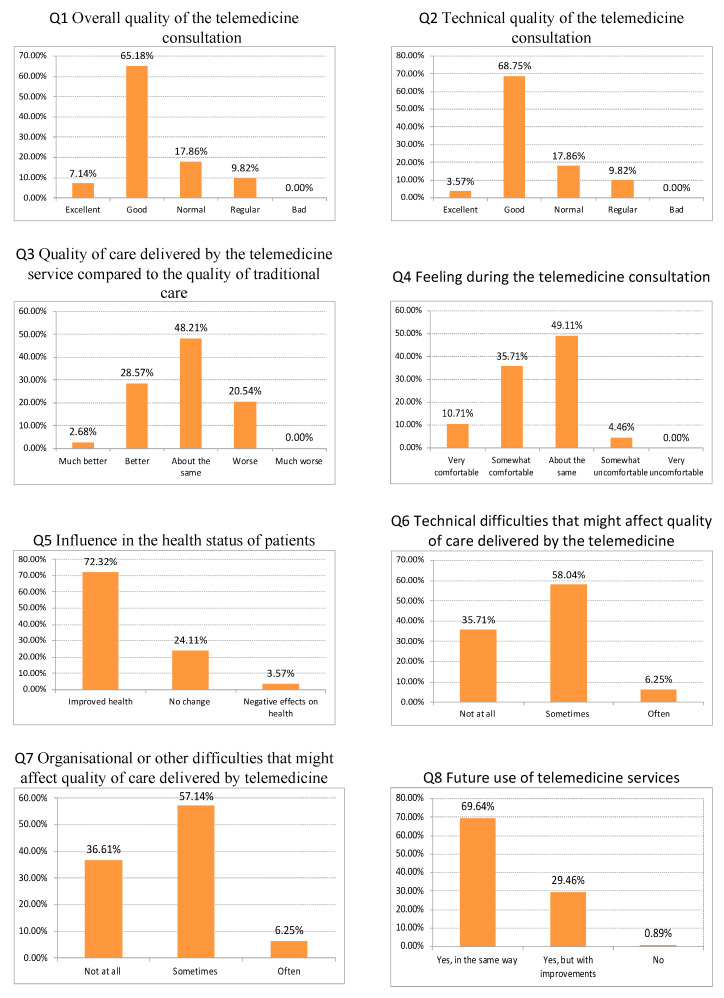
Survey Results.

**Figure 3 ijerph-17-04092-f003:**
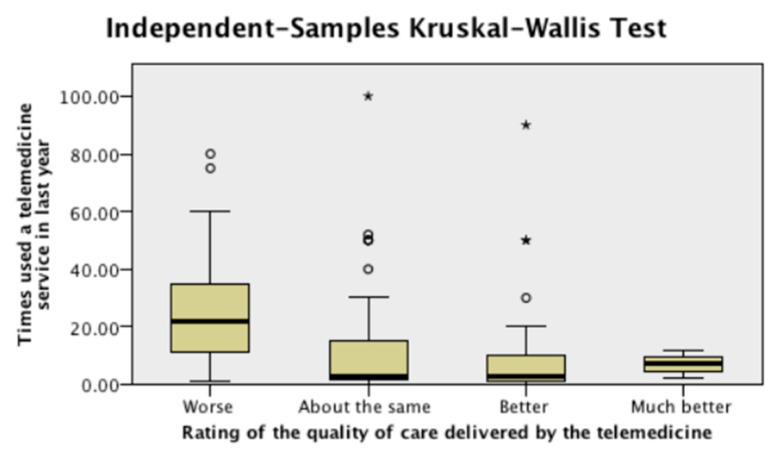
Quality of Care of the Telemedicine Consultation, by Number of Times Used.

**Table 1 ijerph-17-04092-t001:** Sample Characteristics.

Demographics		N
Women n (%)		83 (76.85)
Age (Mean, SD, min–max)		48.4 (9.1) (26–64)
Times used (Mean, SD, min–max)		14.02 (20.82) (1–100)
Professional role n (%)	Physicians	54 (50)
	Nurses	52 (48.15)
	Other	2 (1.85)
Telemedicine services used n (%) (respondents could choose more than one option)	Teledermatology	85 (52.15)
	Teleulcers	64 (39.26)
	Teleaudiometry	46 (28.22)
	Other	14 (8.59)

**Table 2 ijerph-17-04092-t002:** Main Survey Results.

Question	Mean	SD
Q1 How would do you rate the overall quality of the telemedicine consultation? *	3.70	0.74
Q2 How would you rate the technical quality of the telemedicine consultation? *	3.66	0.70
Q3 How would do you rate the quality of care delivered by the telemedicine service when compared to the quality of traditional care? *	3.13	0.76
Q4 Did you feel comfortable during the telemedicine consultation? *	3.53	0.74
Q5 Do you feel that the telemedicine consultation service may influence your patients’ health status? **	2.69	0.54
Q6 Did you experience technical difficulties that might affect the quality of care delivered by the telemedicine service? **	2.29	0.58
Q7 Did you experience organisational or other difficulties that might affect the quality of care delivered by the telemedicine service? **	2.30	0.58
Q8 Would you continue to use the telemedicine service? **	2.69	0.48

* Q1 to Q4 are 5-point Likert whilst ** Q5 to Q8 are 1–3.

**Table 3 ijerph-17-04092-t003:** Differences in Responses by Professional Categories.

Rating the Quality of Care Delivered by the Telemedicine						
Professional category	About the same	Better	Much better	Worse	Much Worse	Total
Medical staff	26	7	2	19	0	54
Nursing staff	24	24	1	3	0	52
TOTAL	50	31	3	22	0	106
Technical difficulties that might affect the quality of care						
Professional category	Often	Sometimes	Not at all	Total		
Medical staff	5	39	10	54		
Nursing staff	2	24	26	52		
TOTAL	7	63	36	106		
Organizational or other difficulties that might affect the quality of care						
Professional category	Often	Sometimes	Not at all	Total		
Medical staff	5	39	10	54		
Nursing staff	2	22	28	52		
TOTAL	7	61	38	106		

**Table 4 ijerph-17-04092-t004:** Linear Correlations between Q1–Q8.

		Rating of The Overall Quality of the Telemedicine Consultation	Rating of the Technical Quality of the Telemedicine Consultation	Rating of the Quality of Care Delivered by the Telemedicine	Feeling during the Telemedicine Consultation	Influence in the Health Status of the Patients	Technical Difficulties That Might Affect the Quality of Care Delivered by the Telemedicine Service	Organisational or Other Difficulties That Might Affect the Quality of Care Delivered by the Telemedicine Service	Future Use of Telemedicine Services
Rating of the overall quality of the telemedicine consultation	Pearson Correlation	1	0.728 **	0.388 **	0.468 **	0.346 **	−0.271 **	−0.422 **	0.583 **
	Sig. (2-tailed)		0	0	0	0	0.004	0	0
	N	112	112	112	112	112	112	112	112
Rating of the technical quality of the telemedicine consultation	Pearson Correlation	0.728 **	1	0.419 **	0.480 **	0.384 **	−0.379 **	−0.363 **	0.505 **
	Sig. (2-tailed)	0		0	0	0	0	0	0
	N	112	112	112	112	112	112	112	112
Rating of the quality of care delivered by the telemedicine	Pearson Correlation	0.388 **	0.419 **	1	0.301 **	0.322 **	0.256 **	−0.353 **	0.308 **
	Sig. (2-tailed)	0	0		0.001	0.001	0.007	0	0.001
	N	112	112	112	112	112	112	112	112
Feeling during the telemedicine consultation	Pearson Correlation	0.468 **	0.480 **	0.301 **	1	0.324 **	−0.054	−0.312 **	0.384 **
	Sig. (2-tailed)	0	0	0.001		0	0.569	0.001	0
	N	112	112	112	112	112	112	112	112
Influence in the health status of the patients	Pearson Correlation	0.346 **	0.384 **	0.322 **	0.324 **	1	−0.183	−0.162	0.348 **
	Sig. (2-tailed)	0	0	0.001	0		0.054	0.088	0
	N	112	112	112	112	112	112	112	112
Technical difficulties that might affect the quality of care delivered by the telemedicine service	Pearson Correlation	−0.271 **	−0.379 **	−0.256 **	−0.054	−0.183	1	0.506 **	−0.331 **
	Sig. (2-tailed)	0.004	0	0.007	0.569	0.054		0	0
	N	112	112	112	112	112	112	112	112
Organisational or other difficulties that might affect the quality of care delivered by the telemedicine service	Pearson Correlation	−0.422 **	−0.363 **	−0.353 **	−0.312 **	−0.162	0.506 **	1	−0.531 **
	Sig. (2-tailed)	0	0	0	0.001	0.088	0		0
	N	112	112	112	112	112	112	112	112
Future use of telemedicine services	Pearson Correlation	0.583 **	0.505 **	0.308**	0.384**	0.348**	−0.331 **	−0.531 **	1
	Sig. (2-tailed)	0	0	0.001	0	0	0	0	
	N	112	112	112	112	112	112	112	112

** Correlation is significant at the 0.01 level (2-tailed); For correlations, we used the 112 participants who answered this part of the questionnaire.

**Table 5 ijerph-17-04092-t005:** Multivariate Linear Regression.

Model	Unstandardized Coefficients		Standardized Coefficients	t	Sig.
	B	Std. Error	Beta		
(Constant)	1.817	0.336		5.404	0.000
Rating of the overall quality of the telemedicine consultation	0.209	0.072	0.322	2.918	0.004
Organizational and other difficulties that might affect the quality of care delivered by the telemedicine service	−0.268	0.077	−0.322	−3.495	0.001

Dependent variable: Q8 Would you continue to use the telemedicine service?

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
