# Peer review of "Primary Care Professionals’ Acceptance of Medical Record-Based, Store and Forward Provider-to-Provider Telemedicine in Catalonia: Results of a Web-Based Survey"

_ijerph, 2020, doi:10.3390/ijerph17114092_

Round 1

Reviewer 1 Report

This is a cross-sectional survey on an important and timely issue pertaining to the adoption of telemedicine among medical professionals.  The presentation is very good and adequately reflecting the state of science in the empirical study based on TAM perspective.  Although it reports interesting results, the paper could be further strengthened in the following areas:

  1. The measurement issues of several basic constructs in the TAM perspective need to be discussed and elaborated. For example, the psychometric properties (validity and reliability) of the scales used should be presented. Because the data are based on self-reported measurements, they are potentially highly correlated.
  2. The association between the two scales (perceived quality of telehealth rendered and perceived barriers of use) should be reported.
  3. A confounder such as the level of telemedicine use is an important variable that should be discussed and considered as a control variable.
  4. The limitations of the study should include the discussion of several measurement issues as well as the confirmation from other perspectives such as users/patients or healthcare administrators.
  5. Future research on the cost-effectiveness issue should be noted.

Author Response

Dear reviewer. Thank you very much for your comments. Please find as follows our point-by-point answers to your suggestions. Please note that line numbers may vary depending on your word editor.

Comment 1. The measurement issues of several basic constructs in the TAM perspective need to be discussed and elaborated. For example, the psychometric properties (validity and reliability) of the scales used should be presented. Because the data are based on self-reported measurements, they are potentially highly correlated.

Response to comment 1: The TAM is a widely renowned and used appraisal method. Our new short-version questionnaire, already referred to in the main text, was validated in a previously published study, showing high reliability index (lines 81-83). The fact that self reporting might imply high correlated data is now mentioned in the discussion.

Comment 2. The association between the two scales (perceived quality of telehealth rendered and perceived barriers of use) should be reported.

Response to comment 2: This is a good point. We looked at it and found no correlation between both variables (Table 4). We have added this in the text (lines 247-248).

Comment 3. A confounder such as the level of telemedicine use is an important variable that should be discussed and considered as a control variable.

Response to comment 3: 

(Lines 231-237) We have looked at differences in responses to the 8 questions of the questionnaire in relation to the number of times respondents have used any of the different telemedicine services in the previous year and we have found significant statistical differences in responses to three questions (we have used the Kruskal-Wallis test instead of the ANOVA test because we found that the variable times used telemedicine in the last year is not a normally distributed variable):

  •   We found significant differences (p=0,05) in the answers to question 3 relating to the rating of the quality of care delivered by the telemedicine services. Respondents who used more often telemedicine services rated the quality of care of these services significantly worse compared with respondents using them less often.
  •   We found significant differences (p<0,001) in the answers to question 6 relating to technical difficulties that might have affected the quality of care delivered by the telemedicine services. Respondents who used more often the telemedicine services stated having experienced more technical difficulties compared with respondents using them less often.
  •   We found significant differences (p<0,001) in the answers to question 7 relating to organisational and other difficulties that might have affected the quality of care delivered by the telemedicine services. Respondents who used more often the telemedicine services stated having experienced more organisational and other difficulties compared with respondents that used telemedicine less often.

Comment 4. The limitations of the study should include the discussion of several measurement issues as well as the confirmation from other perspectives such as users/patients or healthcare administrators.

Response to comment 4:

Unfortunately, the patient and healthcare administrators' perspectives are not assessed in this study, although they are discussed in the introduction of the article. We have mentioned this limitation in section 4 (lines 319-321). 

Comment 5. Future research on the cost-effectiveness issue should be noted.

Response to comment 5: 2 years ago, we conducted a cost-saving analysis, mentioned in the introduction (reference 1). Recently we conducted a cost-minimization analysis, which is now mentioned as well in the introduction (reference 2. Line 49 and Line 74)

Reviewer 2 Report

Review for Manuscript

The Title: Public primary care professionals acceptance of medical record-based, store and forward provider-to-provider telemedicine in Catalonia: results of a validated web-based survey

This manuscripts ’topic is very interesting and deals with important issues according to social issues, also the content of research methods and results is very convincing.

This manuscripts ’topic is very interesting and deals with important issues according to social issues, also the content of research methods and results is very convincing.

Thank you for giving me the opportunity to review these papers.

And in each paragraph, I tell you what you need to modify or consider.

To Authors

Review

Contents

1. Title

It is necessary to summarize the contents of the text through the title so that it can be understood more concisely.

2. Abstract

It is necessary to add more research background of the initial stage.

3. Introduction

It seems that the current status of telemedicine, the strengths and problems of telemedicine should be presented in additional detail. Further explanation is needed in relation to Catalonia's medical and environmental characteristics and telemedicine.

4. Method

- Set the participant flow chart in detail,

- A detailed explanation of the research method is needed.

- It is necessary to explain what was the problem with the questionnaire collection process. 661 to 163 responses to the initial questionnaire transfer, and supplementary explanations to the final 108 respondents' process.

- I'm curious whether the 8 questions included enough to analyze and understand the problems in this study.

5. Data collection

  & Data analysis

- Briefly and accurately explain the analysis process and results, particularly general characteristics

- Please check the number of final participants and the consistency

- I'm curious whether the 8 questions included enough to analyze and understand the problems in this study.

- Present a table appropriate to the journal form

6. Discussion

In this discussion, there was no discussion about the support of the preceding study and the opposite results. Further discussion is required.

7. Conclusion

The conclusions need to be explained in more detail and include suggestions.

Author Response

Dear reviewer. Thank you very much for your comments. Please find as follows our point-by-point answers to your suggestions. Please note that line numbers may vary depending on your word editor.

Comment 1. Title: It is necessary to summarize the contents of the text through the title so that it can be understood more concisely.

Response to comment 1: The title has now been rephrased.

Comment 2. Abstract: It is necessary to add more research background of the initial stage.

Response to comment 2: We have added additional background in the abstract. Lines 24-26 and 39-40.  

Comment 3. Introduction: a) It seems that the current status of telemedicine, the strengths and problems of telemedicine should be presented in additional detail. b)Further explanation is needed in relation to Catalonia's medical and environmental characteristics and telemedicine.

Response to comment 3: 

  1. a) The state of the art of the different impacts of telemedicine (clinical, economic, environmental... emphasizing those related to patient and professionals’ acceptability) is discussed in the first paragraph (lines 48-64).
  2. b) As you suggested, a clarification has been now added to the text (second paragraph) with an additional reference (23). Lines 65-69).

Comment 4. Method: a) Set the participant flow chart in detail, b) A detailed explanation of the research method is needed. c) It is necessary to explain what was the problem with the questionnaire collection process. 661 to 163 responses to the initial questionnaire transfer, and supplementary explanations to the final 108 respondents' process. d) I'm curious whether the 8 questions included enough to analyze and understand the problems in this study.

Response to comment 4:

  1. a) There was an error in the participant flow chart and the numbers were not visible. We have amended it. Thank you for telling us.
  2. b) The first paragraph in the Methods section has a detailed explanation: “This is an observational cross-sectional study using the Catalan version of the Health Optimum questionnaire, a Technology Acceptance Model-based validated survey aimed at measuring the degree of satisfaction with telemedicine services and describing the factors that can determine its future use”
  3. c) There wasn’t any problem with the questionnaire collection process. Our study population of interest was 661 health professionals. Among them, 163 answered the survey, which is a 24,7% response rate. This is quite high as not all 661 professionals use telemedicine.
  4. d) Yes, we can understand your curiosity. However, the aim of this study was precisely to simplify the conventional long and costly TAM questionnaires. Our new questionnaire was validated in a previously published study, showing high reliability index. We have now reinforced this argument at the beginning of the discussion section (lines 264-266 and 309-312).

Comment 5. Data collection & Data analysis: a) Briefly and accurately explain the analysis process and results, particularly general characteristics, b) Please check the number of final participants and the consistency; c) Present a table appropriate to the journal form

Response to comment 5:

  1. We have explained sample general characteristics in Table 1.
  2. We have checked the number of final participants for consistency. Please note that in table 2, we have not included 2 participants that were not medical or nursing staff. In table 4, there are the 112 participants who answered this part of the questionnaire as we didn’t want to bias correlations removing participants who answered this part of the questionnaire. We have now mentioned this as a footnote (line 251).
  3. The tables have now been adjusted to the journal form but we are not sure we have exactly the same format. Please note that the tables are fully editable. Table 4 is quite complex and if the paper is accepted we will need to talk to the editor about putting it as a file.

Comment 6. Discussion: In this discussion, there was no discussion about the support of the preceding study and the opposite results. Further discussion is required.

Response to comment 6:

The discussion includes the results interpretation in the context of the already published studies. As there is not a simple consensus on the drivers of professional’s acceptance this study adds to the literature the specific case of Catalonia. Discussion has been expanded.

Comment 7. Conclusion: The conclusions need to be explained in more detail and include suggestions.

Response to comment 7: The conclusions suggest to “address the technical and organizational difficulties which professionals have encountered if they (the policymakers) are to ensure the use of the services in the future” and highlight the role of the nursing professionals. In our opinion, those are the two big policymaking lessons. As another reviewer suggested as well we have included suggestions about the need for future studies, comparing our results with other perspectives such as users/patients or healthcare administrators (lines 319-321).

Round 2

Reviewer 1 Report

The revised version is responsive to the critique.  I recommend that the paper be accepted for publication.